# Characterization of the First Complete Mitochondrial Genome of Cyphonocerinae (Coleoptera: Lampyridae) with Implications for Phylogeny and Evolution of Fireflies

**DOI:** 10.3390/insects12070570

**Published:** 2021-06-22

**Authors:** Xueying Ge, Lilan Yuan, Ya Kang, Tong Liu, Haoyu Liu, Yuxia Yang

**Affiliations:** 1The Key Laboratory of Zoological Systematics and Application, School of Life Science, Institute of Life Science and Green Development, Hebei University, Baoding 071002, China; gexueying@stumail.hbu.edu.cn (X.G.); 201972421@yangtzeu.edu.cn (L.Y.); 20208017081@stumail.hbu.edu.cn (Y.K.); liutong@stumail.hbu.edu.cn (T.L.); 2College of Agriculture, Yangtze University, Jingzhou 434025, China

**Keywords:** Lampyridae, Cyphonocerinae, *Cyphonocerus sanguineus klapperichi*, mitochondrial genome, characterization, comparative analysis, phylogeny

## Abstract

**Simple Summary:**

The classification of Lampyridae has been extensively debated. Although some recent efforts have provided deeper insight into it, few genes have been analyzed for Cyphonocerinae in the molecular phylogenies, which undoubtedly influence elucidating the relationships of fireflies. In this study, we generated the first complete mitochondrial genome for Cyphonocerinae, with *Cyphonocerus sanguineus klapperichi* as the representative species. The comparative analyses of the mitogenomes were made between *C**. sanguineus klapperichi* and that of well-characterized species. The results showed that the mitogenome of Cyphonocerinae was conservative in the organization and characters, compared with all other fireflies. Like most other insects, the *cox1* gene was most converse, and the third codon positions of the protein-coding genes were more rate-heterogeneous than the first and second ones in the fireflies. The phylogenetic analyses suggested that Cyphonocerinae as an independent lineage was more closely related to *Drilaster* (Ototretinae). Nevertheless, more sampler species are needed in the reconstruction of fireflies’ phylogeny to verify this result.

**Abstract:**

Complete mitochondrial genomes are valuable resources for phylogenetics in insects. The Cyphonoceridae represents an important lineage of fireflies. However, no complete mitogenome is available until now. Here, the first complete mitochondrial genome from this subfamily was reported, with *Cyphonocerus sanguineus klapperichi* as a representative. The mitogenome of *C. sanguineus klapperichi* was conserved in the structure and comparable to that of others in size and A+T content. Nucleotide composition was A+T-biased, and all genes exhibited a positive AT-skew and negative GC-skew. Two types of tandem repeat sequence units were present in the control region (136 bp × 2; 171 bp × 2 + 9 bp). For reconstruction of Lampyridae’s phylogeny, three different datasets were analyzed by both maximum likelihood (ML) and Bayesian inference (BI) methods. As a result, the same topology was produced by both ML analysis of 13 protein-coding genes and 2rRNA and BI analysis of 37 genes. The results indicated that Lampyridae, Lampyrinae, Luciolinae (excluding *Emeia*) were monophyletic, but Ototretinae was paraphyletic, of which *Stenocladius* was recovered as the sister taxon to all others, while *Drilaster* was more closely related to Cyphonocerinae; Phturinae + *Emeia* were included in a monophyletic clade, which comprised sister groups with Lampyridae. *Vesta* was deeply rooted in the Luciolinae.

## 1. Introduction

Lampyridae Rafinesque, 1815 is a cosmopolitan family consisting of about 100 genera and 2200 species [1,2,3]. It is an amazing bioluminescent beetle group, with all species found to be luminous, at least at the larval stage [2]. However, some lineages exhibit no or weak bioluminescence in adulthood, leading to considerable confusion in their taxonomic positions. One of such non-luminescent groups is the monotypic subfamily Cyphonocerinae Crowson, 1972, which represents an important lineage with the type of genus *Cyphonocerus* Kiesenwetter, 1879 [3]. The adults of *Cyphonocerus* are easily recognized by the bipectinate antennae [4]. It is a small genus and has 17 species (subspecies) hitherto known from Japan, China, Nepal, and N. India [5]. However, there has been inconsistency in its family-group assignment since its establishment. 

The genus *Cyphonocerus* was originally established under the family Drilidae Blanchard, 1845 [6,7,8,9,10] (now as a tribe of Elateridae Leach, 1825 [11]), and transferred to Lampyridae by Nakane [12]. It was once placed in the subfamily Amydetinae Olivier in Wytsman, 1907 [13] and later as the type genus to establish the subfamily Cyphonocerinae by Crowson [14], which was recognized as a subjective synonym of Psilocladinae McDermott, 1964 by Jeng et al. [4], and synonymized with Lampyrinae by Lawrence et al. [15], but most recently revalidated by Martin et al. [3] with *Cyphonocerus* left as the sole member.

Recent phylogenetic efforts have provided deeper insight into the classification of fireflies by expanding morphological [16,17], or molecular [2,3,18,19], or both datasets [19]. All of these studies highlighted the need to update the higher-level classification of Lampyridae within a phylogenetic framework. However, in the previous molecular phylogenetic studies, none [3,19] or limited genes [20,21] have been analyzed for Cyphonocerinae, or not all sampler species were provided with the same number of gene markers in reconstructing phylogenetic trees [2], this undoubtedly influenced elucidating the relationships of fireflies. Until now, only *16S*, *cox1*, *nad5*, *18S* genes are available for *Cyphonocerus* in the public database [21,22,23], so more thorough gene analyses are needed.

Complete mitochondrial genomes have been widely used in investigating molecular evolution and phylogenetic relationships among different lineages of insects due to their highly conserved structure in evolution, rare recombination, and rapid evolutionary rate [24,25,26]. In mitochondrial genomes, compositional bias and substitution rate variation have been extensively investigated in comparative analyses since that they provide critical information for molecular evolution [27,28,29,30,31]. Furthermore, phylogenomic analysis with higher numbers of genes, up to all of the 37 mitochondrial genes, has been tested to get more highly supported nodal confidence, compared with a single or a few locus phylogenetics [28,32,33].

In this study, we generated and analyzed the first complete mitochondrial genome for *Cyphonocerus sanguineus klapperichi* Pic, 1955. This enabled us to provide the comparative analysis of the genomic structure, base composition, substitution, and evolutionary rates between Cyphonocerinae and that of well-characterized complete mitogenomes of other fireflies, as well as a comprehensive molecular phylogenetic analysis of Lampyridae based on complete mitogenomes. The complete mitogenome reported here will contribute to fireflies’ higher phylogeny reconstruction based on mitogenome sequences and promote comparative mitogenome studies, which should help understand the mitogenome evolution across different lineages of Lampyridae.

## 2. Materials and Methods

### 2.1. Taxon Sampling

The material of *C. sanguineus klapperichi* was collected from China, Fujian Province, Wuyishan, Tongmu, San’gang, 117°41′16′′ E, 27°45′10′′ N, on 23 May 2018. Specimens were preserved in 100% ethanol at ‒20 °C before molecular experiments. The analyzed specimen was identified using the identification key provided by Jeng et al. [5].

### 2.2. DNA Extraction, Mitochondrial Genome Sequencing, and Assembly

Total genomic DNAs were extracted using a DNeasy Blood & Tissue kit (QIAGEN, Beijing, China), according to the manufacturer’s instructions. DNA was stored at −20 °C for long-term storage and further molecular analyses, which were deposited in the Museum of Hebei University (MHBU, accession No. 2CAN0196).

The whole mitochondrial genome sequence was sequenced using an Illumina Novaseq 6000 platform with 150 bp paired-end reads at BerryGenomics, China. The sequence reads were first filtered by the programs following Zhou et al. [34], and then the remaining high-quality reads were assembled using IDBA-UD [35], under similarity threshold 98%, and k values minimum 40 and maximum 160 bp. The gene *cox**1* was amplified by polymerase chain reaction (PCR) using universal primers as ‘reference sequences’ to target mitochondrial scaffolds by IDBA-UD [35] to acquire the best-fit, which is under at least 98% similarity. Geneious 2019.2 [36] software was used to manually map the clean readings to the obtained mitochondrial scaffolds to check the accuracy of the assembly.

### 2.3. Genome Annotation and Analyses

Gene annotation was done by Geneious 2019.2 [36] software and the MITOS web server (http://mitos.bioinf.uni-leipzig.de/index.py, accessed on 20 March 2021) [37]. The positions and secondary structures of 22 tRNAs were estimated by a combination of the results predicted by an ARWEN and tRNAscan-SE Search Server v.1.21 [38,39]. The mitogenomic circular map was produced using a CGView Server (http://stothard.afns.ualberta.ca/cgview_server, accessed on 20 March 2021) [40]. The skewness was determined with base composition of nucleotide sequences by using the formula: AT skew = [A − T]/[A + T], GC skew = [G − C]/[G + C] [41]. The Tandem Repeat Finder program was used to predicted tandem repeats in A + T-rich region [42]. The relative synonymous codon usage (RSCU) was analyzed by MEGA 7.0 [43]. DnaSP v5.10.01 [44] was used to calculate the nucleotide diversity (Pi) and sliding window analysis (a sliding window of 200 bp and a step size of 20 bp) based on 13 aligned protein-coding genes (PCGs) and non-synonymous (Ka)/synonymous (Ks) substitution rates among the 13 PCG. The base composition and component skew were analyzed using PhyloSuite v1.2.2 [45]. The genetic distances were computed using MEGA 7.0 with the Kimura-2-parameter model. SymTest v2.0.47 [46] with Bowker’s matching pair symmetry test was used to analyze the differences of heterogeneous sequences in the datasets, and the heat maps were generated according to the inferred *p*-values.

### 2.4. Phylogenetic Analysis

We followed the classification of Lampyridae by Martin et al. [3]. In addition to the newly sequenced genome here for Cyphonocerinae, another 32 species representing four subfamilies of Lampyridae were selected as the ingroups, which are the previously published complete or almost complete mitochondrial genomes downloaded from GenBank (Table 1). Two species of Rhagophthalmidae (*Rhagophthalmus ohbai* Wittmer, 1994) and Phengodidae (*Phrixothrix hirtus* Olivier, 1909) were chosen as the outgroups [2].

Data standardization and information extraction were performed by PhyloSuite v 1.2.2 [45]. The 13 PCGs were aligned using the MAFFT algorithm implemented in TranslatorX [47] with the L-INS-i strategy. The 2 rRNAs and 22 tRNAs were aligned with MAFFT version 7 online services using the G-INS-i strategy. Gblocks v 0.91b [48] was used to remove the gaps and ambiguously aligned sites. The aligned data were concatenated with Sequence Matrix v.1.7.8 [49] and PhyloSuite v 1.2.2. The alignment of the individual gene was concatenated into three datasets: (i) the PCGrRNA matrix, including 13 PCGs and 2 rRNA genes (12,875 bp), (ii) the PCG12RNA matrix, including the first and second codon positions of the PCGs and 2 rRNA genes (9236 bp), and (iii) the PCGRNA matrix, including 13 PCGs, 2 rRNA genes and 22 tRNA genes (14,321 bp).

All datasets were analyzed using maximum likelihood (ML) on the IQ-TREE web server (http://iqtree.cibiv.univie.ac.at/, accessed on 6 April 2021) [50] with the GTR+G+I substitution model (Appendix A). Bayesian inference (BI) was also used for the phylogenetic analyses either by PhyloBayes MPI v.1.7a [51] (for the PCG12RNA and PCGrRNA matrixes) with the site-heterogeneous mixture CAT + GTR model (Appendix A), or by MrBayes 3.2.6 [52] (for the PCGRNA matrix) with two independent Markov Chain Monte Carlo chains run of 2 × 10^6^ generations, of which the tree was sampled every 1000 generations, and the initial 25% of sampled data were discarded as burn. ITOL (http://itol.embl.de/, accessed on 10 April 2021) [53] was used to annotate and beautify the phylogenetic tree.

## 3. Results

### 3.1. Genomic Structure and Base Compositions

As in other fireflies, the mitogenome of *C. sanguineus klapperichi* (Figure 1) is a typical double-strand circular molecule and contains 13 protein-coding genes (PCGs), 22 transfer RNA genes (tRNAs), 2 ribosomal RNA(rRNAs) genes, and a control region (CR) or AT-rich region, in which 14 genes (8 tRNAs, 4 PCGs, and 2 rRNAs) are transcribed from the minority strand (N-strand), while others (14 tRNAs and 9 PCGs) from the majority strand (J-strand). The annotated sequence was registered in GenBank with accession number MW365445. This is the first complete mitogenome record for Cyphonocerinae.

Seven gene overlaps are present in *C. sanguineus klapperichi* mitogenome (Appendix A), ranging from 1 to 4 bp, with the longest overlap (4 bp) occurring between *atp6* and *atp8*, and also *nad4* and *nad4l*, respectively. The overlap between *atp6* and *atp8* is also found in mitogenomes of other arthropods [66,67,68]. Moreover, there are 13 intergenic spacer regions between genes (Appendix A), of which the total length is 211 bp, with the longest intergenic spacer (72 bp) exists between *trnC* and *trnW*. This result shows that the number and length of gene spacers are significantly higher than those of gene overlaps.

Known complete mitogenomes of fireflies range from 15,950 bp (*Asymmetricata circumdata*) to 18,054 bp (*Pyrocoelia thibetana*) (Appendix A). The mitogenome of *C. sanguineus klapperichi* is 16,443 bp in length (Appendix A) and slightly shorter than most others, of which the average length is 16,855 bp.

For fireflies’ mitogenomes, the sizes of the control region vary greatly among different species, whereas the PCGs, tRNAs, and rRNAs show little variation in length (Figure 2A, Appendix A). This suggests that the mitogenome size of different fireflies is largely determined by the size of control regions, like other insects [68].

The base composition of *C. sanguineus klapperichi* is A (42.5%), T (34.2%), C (13.9%), and G (9.4%), respectively (Table 2). It contains a slightly lower A+T content (76.7%) and a higher G + C content (23.3%), compared to that of other firefly species, which have an average value of A + T content being 78.0%, varying from 75.7% to 80.7% (Appendix A). This lower A + T bias in *C. sanguineus klapperichi* is reflected in all components of its genome, except tRNAs are near to the average value (Figure 2B).

The nucleotide skew analysis shows that the full mitogenome of *C. sanguineus klapperichi* exhibits a positive AT-skew (0.11) and a negative GC-skew (−0.19) (Table 2). A similar pattern is found in all other firefly mitogenomes, with the AT-skew ranging from 0.06 (*Photinus pyralis*) to 0.14 (*Luciola cruciata*) and a GC-skew varying from −0.08 (*Diaphanes nubilus*) to −0.24 (*Lamprigera yunnana*) (Appendix A). These results indicate that Ototretinae (*Drilaster* sp. and *Stenocladius* sp.) has the strongest A skew and weakest C skew, while Cyphonocerinae (*C. sanguineus klapperichi*) has an average value of AT-skew and GC-skew in comparison with other known firefly mitogenomes (Figure 2C,D). This base composition bias has been suggested to be associated with replication and transcription of the mitochondrial genome [67]. 

### 3.2. Protein-Coding Genes

The overall size (excluding stop codons) of 13 PCGs of *C. sanguineus klapperichi* is 11,008 bp in length, accounting for 66.95% of the total genome (Table 2). Like the full mitogenome, the whole PCGs show a slightly lower A+T content (75.1%), of which the third codon position (80.2%) is higher than those of the first and second codon positions (72.3% and 72.9%, respectively). The AT-skew (0.09) is positive, while GC-skew (−0.17) is negative for the PCGs, reflecting a bias towards nucleotides A and C than their counterparts.

All PCGs of *C. sanguineus klapperichi* are initiated with the standard ATN codons and terminated with TAA/TAG or a truncated termination codon T (Appendix A). These incomplete stop codons are thought to be ubiquitous in metazoan [26] and have been supposed to be completed through posttranscriptional polyadenylation [69].

The codon usage analysis of *C. sanguineus klapperichi* shows that the most frequently used codons are UUA-Leu (352), AUU-Ile (346), UUU-Phe (326), and AUA-Met (235) (Appendix A, Appendix A). The UUA-Leu also has the highest RSCU value (3.9), further indicating that UUA is the most preferred codon. The RSCU values of the PCGs reveal that there is a higher frequency in the usage of AT than that of GC in the third codon positions (Appendix A, Appendix A).

Sliding window analysis was implemented to study the nucleotide diversity of 13 PCGs among fireflies exhibited in Figure 3A. Nucleotide diversity values range from 0.187 (*cox1*) to 0.337 (*atp8*). Among the genes, *atp8* (Pi = 0.337) has the highest variability, followed by *nad6* (Pi = 0.303), *nad2* (Pi = 0.282), and *nad3* (Pi = 0.257) (Appendix A). In contrast, *cox1* (Pi = 0.187) and *nad1* (Pi = 0.198) have relatively low values and are the most conserved of the 13 PCGs. This result indicates that the nucleotide diversity is highly variable among the 13 PCGs.

Pairwise genetic distances among the mitogenomes of fireflies (Figure 3B, Appendix A) show that *atp8* (0.439), *nad6* (0.423), and *nad2* (0.360) evolve comparatively faster, while *cox1* (0.216) and *nad**1* (0.234) evolve comparatively slowly.

The ratio (ω) of non-synonymous (Ka) to synonymous (Ks) substitution rates, which is a diagnostic statistical method to detect molecular adaption [70,71], is used to estimate the evolutionary rate among insects. In Lampyridae, the genes *atp8* (0.777), *nad6* (0.641), and *nad2* (0.501) have comparatively high Ka/Ks ratios, while *cox1* (0.165), *cox2* (0.266), and *cox3* (0.282) have relatively low values (Figure 3B, Appendix A). The average Ka/Ks (ω) of 13 PCGs of the fireflies are all less than 1, indicating that these genes are under purifying selection [72].

Heterogeneity of nucleotide divergence was examined under pairwise comparisons in a multiple sequence alignment (Figure 4). The datasets PCGrRNA and PCGRNA exhibit higher heterogeneous sequence divergence than PCG12rRNA, indicating that the third codon positions are more rate-heterogeneous than the first and second ones. The higher compositional heterogeneity may result in systematic errors in phylogenetic analyses [27].

### 3.3. Transfer and Ribosomal RNA Genes

The complete set of 22 typical tRNAs were all found in the mitochondrial genome of *C. sanguineus klapperichi*, and their secondary structures are shown in Appendix A. The total length of tRNAs is 1432 bp, ranging from 62 bp (*trnF*, *trnG*, *trnH,* and *trnL1*) to 71 bp (*trnK*) in size (Appendix A). The AT-skew (0.07) is positive, and GC-skew (−0.15) is negative (Table 2), indicating a preference for using A base over T and G over C.

Most tRNAs exhibit the typical clover-leaf structures, except *trnS1* missing the dihydrouridine (DHU) arm (Appendix A). Lacking the DHU arm in *trnS1* is a common feature for most metazoan mitogenomes [26]. These aberrant tRNAs are supposed to sustain their function via a posttranscriptional RNA editing mechanism [73,74]. In the tRNAs, except the classic base pairs (A-U and C-G), 22 non-canonical base pairings (G-U and A-C), and 8 other mismatched base pairs (U-U, A-A, A-G) were found in the arms (Appendix A).

In the mitogenome of *C. sanguineus klapperichi*, the *rrnL* (1268 bp) and *rrnS* (766 bp) are located in the conserved positions between *trnL* and *trnV*, and *trnV* and the control region, respectively (Figure 1). There is a little variation in the sizes of both rRNAs among firefly species (Figure 2A, Appendix A). The A+T content of *rrnL* and *rrnS* are 81.5% and 78.7%, respectively (Appendix A), which are slightly lower than most other firefly species (Figure 2B). The overall rRNA shows a positive AT-skew (0.17) and a negative GC-skew (−0.34), which shows a slight bias toward using A and an obvious bias toward G (Table 2).

### 3.4. Control Region

The control region of *C. sanguineus klapperichi* was identified by the position between *rrnS* and *trnI*, spanning 1776 bp in length (Figure 1, Table 2). This is comparable to that of most other fireflies, which have an average length of 1810 bp (Appendix A), ranging from 1400 bp (*Photinus pyralis*) to 2341 bp (*Diaphanes* sp.). Generally, the length of the control region varies more than other components in the fireflies (Figure 2A). The control region is supposed to be involved in the initiation of replication and transcription of mitogenomes [75].

The control region of mitogenome in *C. sanguineus klapperichi* has a slightly lower A+T content (81.2%) compared to most other firefly species (Figure 2B), of which the average value is 86.6% (Appendix A). The AT-skew (0.12) is positive, while the GC-skew (−0.23) is negative (Table 2), showing a bias towards using A and G.

Within the control region, two tandem repeat sequence units are present in the mitogenome of *C. sanguineus klapperichi*; their positions and length are shown in Appendix A. They are a 136 bp-sequence tandemly repeated twice and a 171 bp-sequence tandemly repeated twice with a partial third repeat (9 bp), respectively. In Lampyridae, the tandem repeat sequences within the control region are quite diverse. They have been found in most firefly species which are all located between *rrnS* and *trnI* and vary widely in size and number of repeat units but are absent in *Luciola substriata* (Appendix A). The species has two different types of repeat units at least, and every unit repeated at least twice, except *Ellychnia corrusca*, *Bicellonycha lividipennis,* and *Asymmetricata circumdata*, which have only one type of repeat unit. The most diverse tandem repeat sequence units happen in *Pyrocoelia praetexta*, which includes five types of repeat units. The longest repeat unit was found in *Diaphanes mendax*, which contained two 303 bp repeats. However, in *Diaphanes pectinealis*, the tandem repeat region is the shortest, with a 9 bp repeat unit tandemly repeated six times plus a 6 bp partial sequence.

### 3.5. Phylogenetic Analysis

Analyses of the three datasets resulted in nearly identical and fully resolved topologies with high nodal support values under ML and BI methods. What is noted, the BI reconstruction of 37 genes dataset and ML reconstruction of the 13 PCGs and 2rRNA dataset produced the same topology shown in Figure 5.

In all topologies (Figure 5, Appendix A), the monophyly of Lampyridae was well supported based on mitogenomes of different genes datasets (PP = 1/0.98/0.92, BS = 100/100/96). Furthermore, the monophyly of Lampyrinae and Luciolinae Lacordaire, 1857 (with *Emeia* Fu, Ballantyne *et* Lambkin, 2012 excluded), including 11 and 16 representative species respectively, were highly supported (PP = 1, BS = 100). The clade composing Photurinae Lacordaire, 1857 (only one species included) and *Emeia* was suggested to be a monophyly (PP = 1, BS = 100), which was a recovered sister to Lampyrinae with high support value (PP = 1, BS = 100). However, the monophyly of Ototretinae McDermott, 1964 was not recovered, with the sampler genera *Drilaster* Kiesenwetter, 1879 and *Stenocladius* Fairmaire *in* Deyrolle and Fairmaire, 1878 splitting in different clades. The monophyly of the monotypic Cyphonocerinae with a single species here could not be tested.

Except for the ML reconstruction of PCGRNA dataset (Appendix A), *Stenocladius* was recovered as the sister taxon to all other fireflies, albeit with comparatively high support values (PP = 0.854/0.74/0.59, BS = 90/94).

In the topologies produced by ML analyses of all datasets (Figure 5, Appendix A) and BI analysis of PCGRNA dataset (Figure 5), *C. sanguineus klapperichi* (Cyphonocerinae) was always grouped with *Drilaster* (Ototretinae), with a high support value (BS = 98/93/83; PP = 1); then except ML analysis of PCGRNA dataset, which together with a sister to Lampyrinae + (Photurinae + *Emeia*) (Figure 5, Appendix A), was highly supported (BS = 76/81; PP = 0.924), while in the latter (Appendix A), they were a recovered sister to Luciolinae but with very low support (BS = 35). However, based on the other two datasets of BI analyses (Appendix A), *C. sanguineus klapperichi* was solely suggested as a sister to Lampyrinae + (Photurinae + *Emeia*), also with high support values (PP = 0.99/0.89), while *Drilaster* was in uncertain relationships with or sister to these taxa.

Based on the BI reconstruction of PCGrRNA and PCG12rRNA datasets (Appendix A) and ML reconstruction of PCG12rRNA dataset (Appendix A), *Lamprigera* Motschulsky, 1853 was a recovered sister to Luciolinae, with comparatively high support values (PP = 0.99/0.99; BS = 63). However, it was a recovered sister to the remaining fireflies except *Stenocladius* under ML analysis of PCGrRNA dataset and BI analysis of PCGRNA dataset (Figure 5), also with high support (BS = 90; PP = 0.854). Moreover, it was possibly a sister to *Stenocladius* based on ML analysis of PCGRNA dataset (Appendix A), but with a low support value (BS = 56).

In all topologies (Figure 5, Appendix A), *Vesta* Laporte, 1833 was deeply rooted in Luciolinae and a sister to *Pristolycus* Gorham, 1883, which was greatly supported (PP = 1, BS = 100). Surprisingly, *Emeia* was always grouped with *Bicellonycha* Motschulsky, 1853 (Photurinae), which was highly supported (PP = 1, BS = 100) in all analyses.

## 4. Discussion

### 4.1. Features of Mitochondrial Genomes in Lampyridae

In all known mitochondrial genomes of Lampyridae, both the size and A+T content vary greatly for the control region, but a less variation for PCGs, tRNAs, and rRNAs, respectively, indicating that the control region is an important component that heavily affects the size and total A+T content of fireflies’ mitogenome.

Fireflies exhibit the typical A+T-biased composition of insect mitogenomes [26,76,77], in either the full genome or each component, all over 75.1%. The biological reasons for such A+T-biased compositional heterogeneity have been extensively investigated [78,79,80], and one of the hypotheses is the energy efficiency trade-offs which has been experimentally rested [79]. The hypothesis suggests that the resources for nucleotide production are limited, and synthesis of G+C consumes more energy and nitrogen than A+T, so A and T are preferred nucleotides [79]. Although the mitogenome of fireflies is A+T-biased, Cyphonocerinae have a slightly lower value of A+T content in comparison with most others (Figure 2A). What is more interesting, the size of the full genome of Cyphonocerinae is about 400 bp shorter than those of the other fireflies (Appendix A). Therefore, it is presumed that the higher G+C content in this group seems to be compensated by the shortened mitogenome, which is likely to be shaped by selection for efficient usage [30]. Additionally, this is suggested to be a molecular strategy to ensure a reliable protein synthesis under high temperatures [76].

In Lampyridae, the AT-skew is all positive while GC-skew is negative, indicating the base composition bias towards A and C than their counterparts. Although the cases for such skewed strand composition are multifactorial, most of the hypotheses suggest that the strand asymmetry is the result of mutations and selection pressures [81], and the value of GC-skew of insect mitogenomes seem to be associated with replication orientation [79].

The analyses of nucleotide diversity, pairwise genetic distances, and Ka/Ks (ω) all showed that in the Lampyridae, the genes *atp8*, *nad6,* and *nad2* to be more variable and evolve faster, while *cox1* is more conserved and evolves comparatively slowly. Nucleotide diversity analyses are useful for designing species-specific markers, especially in taxa where morphological identification is difficult and ambiguous [82,83]. The *cox1* gene is often used as a universal barcode for species identification for the insects [84,85,86], but for the fireflies, its low variability indicates that it is more suitable for exploring the phylogenetic relationships among the higher grades. In contrast, those genes exhibiting an optimal combination of fast evolution and sufficiently large size, such as *nad6*, should be evaluated as potential DNA markers for species and/or population identification.

### 4.2. Phylogenetic Implications of Mitochondrial Genomes in Lampyridae

As a morphologically and biologically diverse group [16,17,18,87], the taxonomy and classification of fireflies have been extensively debated [2,3,7,14,15,20,88,89,90,91], particularly the non-luminescent groups are more controversial, such as Ototretinae and Cyphonocerinae.

The subfamily Ototretinae was established by McDermott [92] and redefined in a broad sense by Crowson [14], which was followed by Lawrence and Newton [91] and Brancucci and Geiser [93]. This group consists of several genera, including *Drilaster* and *Stenocladius*. However, Branham and Wenzel [16] excluded the latter taxa from Lampyridae on the basis of a morphological phylogeny, which was supported by Lawrence et al. [15], who placed them in Elateriformia *incertae sedis*, but against the molecular phylogenetic analyses by Bocakova et al. [22] and Sagegami-Oba et al. [94], also not adopted by Geisthardt and Satô [95]. Recently, this subfamily was revised by Janisova and Bocakova [96], based on the comparative morphology of adults, and transferred the above genera to Lampyridae again. In the most recent molecular phylogenetics, the members of Ototretinae were deeply rooted in Lampyridae with high support value [2,3,20] but often suggested to be a paraphyletic group [2,3], which is congruent with the present study.

Furthermore, our analyses indicated that *Stenocladius* was most probably the basal lineage of Lampyridae, which correlated with that of Li et al. [97]. In addition, *Cyphonocerus* (Cyphonocerinae) was recovered more closely related to *Drilaster*, which is in agreement with that of Martin et al. [20]. The clade of *Drilaster* and *Cyphonocerus* seemed more closely related to Lampyrinae but Luciolinae, which is in agreement with Suzuki [21], but against those results of Martin et al. [20] or Chen et al. [2].

Unfortunately, the monophyly of Cyphonocerinae could not be tested in this study due to a lack of more material of *Cyphonocerus*. Furthermore, the phylogenetic relationships among *Psilocladus* Blanchard, 1846 and *Pollaclasis* Newman, 1838 were not evaluated because of a deficiency of the complete mitogenome data. The three taxa were included in the subfamily Psilocladinae by Jeng [4,5], which was redefined by Martin et al. [3] as a monotypic subfamily (including only *Psilocladus*) and *Pollaclasis* in Lampyridae *incertae sedis*, meanwhile, *Cyphonocerus* left as the sole member of Cyphonocerinae. The efforts need to be made to clarify their relationships in the future when the material is available for all these genera.

Previous works have recovered *Lamprigera* in various positions within Lampyridae [3,20,61,97]. In this study, *Lamprigera* was either a recovered sister to Luciolinae, which is congruent with Martin et al. [3] and Chen et al. [2], or a sister to the remaining fireflies except for *Stenocladius*, similar to that of Li et al. [97], but never be a member of Lampyrinae [98]. Given this incongruence, the exact position of *Lamprigera* remains uncertain, as what has been done by Martin et al. [3,20], placing it as Lampyridae *incertae sedis*. To rigorously test the classification of *Lamprigera* relative to other subfamilies, an expanded taxon sampling including deeper species coverage of this genus will be needed.

Additionally, here we recovered *Vesta* in the Luciolinae for the first time and sister to *Pristolycus* with strong support and congruence among all of our analyses. It was once placed in Amydetinae [92] or Lampyrinae [90] and was noted to be a paraphyletic group with some species from Photurinae and Lampyrinae by Jeng [98]. Evidence from individual or multi-molecular markers supported the position of *Vesta* near Photurinae in the phylogenetics [2,3,20,97] and was placed in the Lampyridae *incertae sedis*. However, this placement was based on a single *Vesta* species; as a specious genus, more taxa should be included in the future study to verify this result.

It is surprising that the monotypic *Emeia* and *Bicellonycha* (Photurinae) comprised a monophyletic clade in all of our analyses. Regardless of this result, their morphological characteristics, biology, and luminous behaviors are substantially different [99]. In addition, its placement in Luciolinae has been well supported by both morphological [99] and molecular phylogenies [2,3]. Since only one species of Photurinae was included in our analyses, an expanded sampling taxon will be needed to test this placement, so we do not make any taxonomic change here.

Above all, the molecular phylogenies, including this study, were analyzed on the basis of a minority part of species or limited molecular data in comparison to an estimated 2200 species of fireflies worldwide. Therefore, many more species need to be included in future analysis to establish a solid and dependable classification of Lampyridae. Particularly, the complete mitochondrial genomes should be encouraged to accumulate more for Lampyridae, in view of its high value in investigating phylogenetic relationships of the insects.

## 5. Conclusions

In the present study, we generated and analyzed the first complete mitochondrial genome for Cyphonocerinae, with *C**. sanguineus klapperichi* as a representative. Compared with that of all well-characterized mitochondrial genomes of fireflies, the mitogenome of Cyphonocerinae is highly conserved in structure and size. It is a highly A+T-biased composition, with a positive AT-skew and negative GC-skew. It has conserved codon usage of protein-coding genes and secondary structures of tRNAs, as well as a unique type of tandem repeat sequence units present in the control region. This provides the basic information to perform comparative analyses and further discussion of the mitogenomes’ evolution of Lampyridae.

Furthermore, the nucleotide diversity, genetic distance substitution rates, and heterogeneity of nucleotide divergence were analyzed and examined. The result indicates that in Lampyridae, the genes *atp8*, *nad6,* and *nad2* are more variable and evolve faster, while *cox1* is more conserved and evolves comparatively slowly. Furthermore, the third codon positions are more rate-heterogeneous than the first and second ones.

Moreover, the phylogenetic trees of Lampyridae were reconstructed based on three different datasets by both maximum likelihood (ML) and Bayesian inference (BI) methods. The result suggests that Lampyridae, Lampyrinae, Luciolinae (excluding *Emeia*) are monophyletic. Ototretinae is paraphyletic, of which *Stenocladius* is at the basal lineage and sister to all others, while *Drilaster* is more closely related to Cyphonocerinae. Lampyridae + (Photurniae + *Emeia*) comprises sister groups, and the latter two are a monophyletic clade. Here *Vesta* is recovered in Luciolinae for the first time. Nevertheless, large-scale analyses with denser taxon sampling are needed to confirm the present results.

## Figures and Tables

**Figure 1 insects-12-00570-f001:**
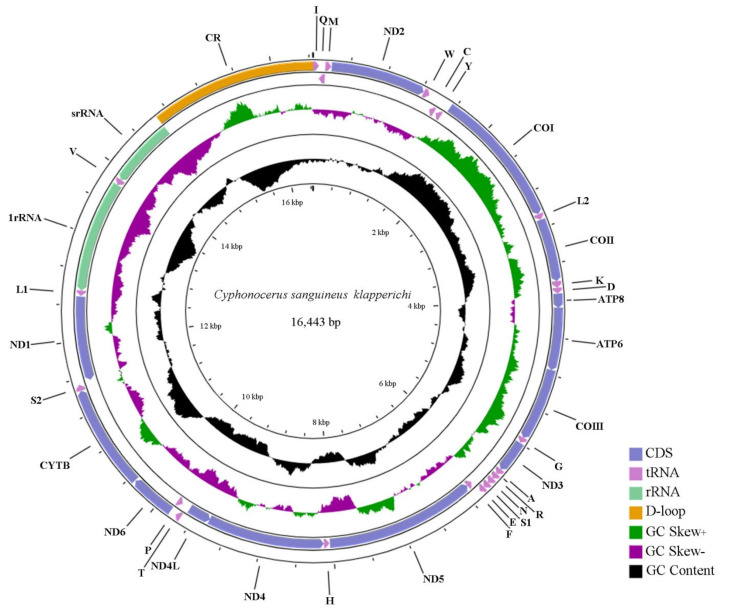
Circle map of the complete mitogenome of *C. sanguineus klapperichi*. Different colors indicate different types of genes and regions. Genes shown at the outer circle are located on the J-strand, and those at the inner circle are located at the N-strand.

**Figure 2 insects-12-00570-f002:**
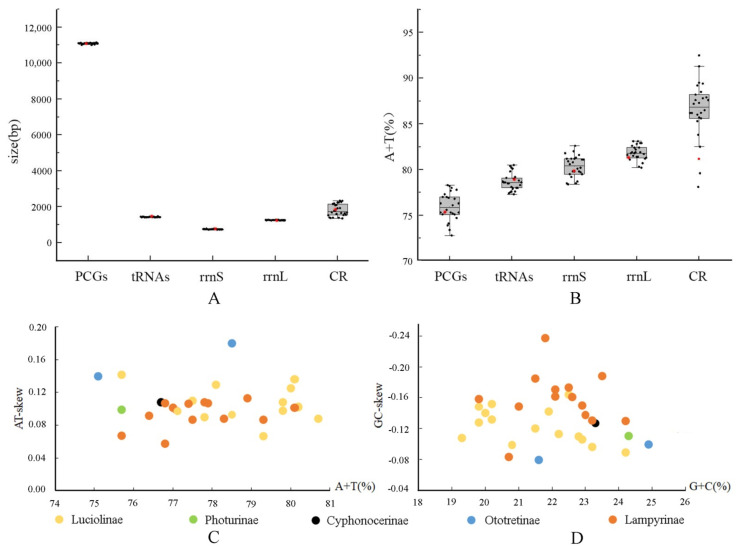
The size (**A**) and AT content (**B**) of PCGS, *rrnL*, *rrnS*, CR, and tRNA of 33 firefly species with the red dots representing *C. sanguineus klapperichi* (lower edge of the gray rectangle, 25 percentile; central black bar within the rectangle, median; upper edge of the rectangle, 75 percentile); Nucleotide composition of 33 complete firefly species, with the black dots representing the *C. sanguineus klapperichi*: (**C**) the A+T content and AT skew; (**D**) the G+C content and GC-skew.

**Figure 3 insects-12-00570-f003:**
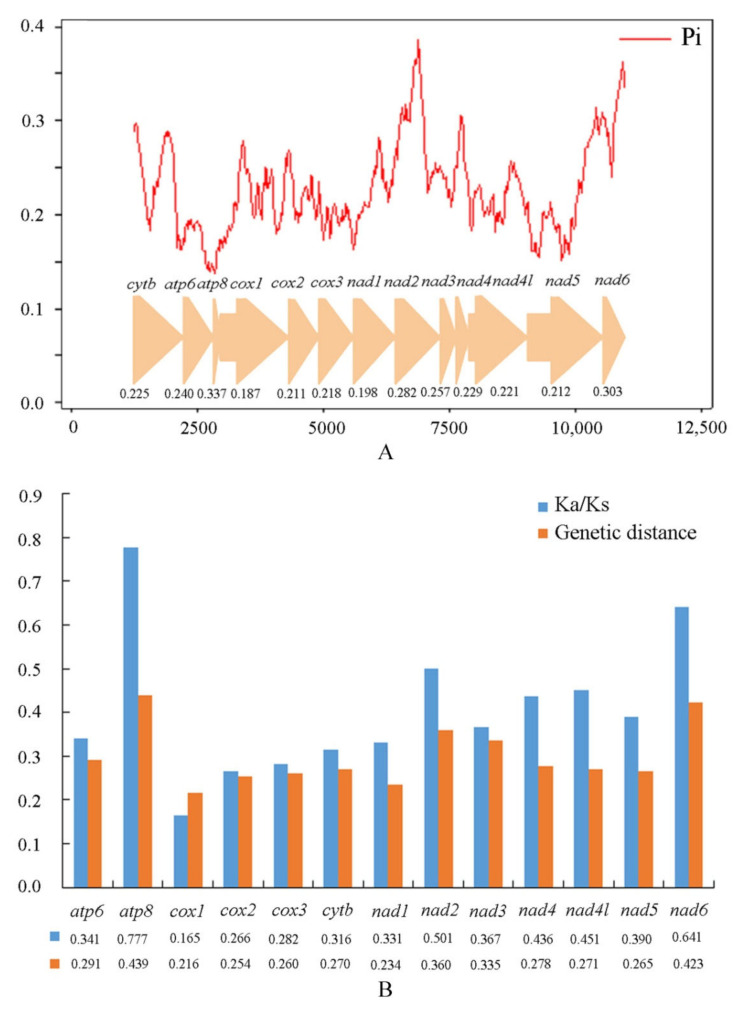
(**A**) The nucleotide diversity (Pi) of 13 protein-coding genes of mitogenome among 33 species of Lampyridae in a sliding window analysis (a sliding window of 200 bp with the step size of 20 bp); the Pi value of each gene is shown under the gene name. (**B**) Genetic distances and the ratio of non-synonymous (Ka) to synonymous (Ks) substitution rates of 13 protein-coding genes among 33 species of Lampyridae. The average value for each PCGs is shown under the gene name.

**Figure 4 insects-12-00570-f004:**
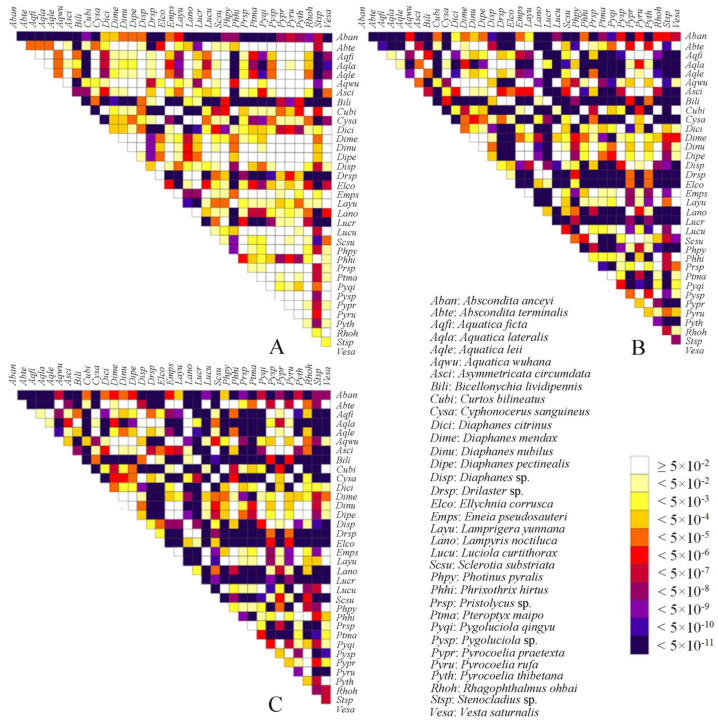
Heterogeneous sequence divergence of mitochondrial genomes of Lampyridae resulting from pairwise comparison of three aligned datasets: (**A**) PCG12rRNA; (**B**) PCGrRNA; (**C**) PCGRNA. The dark colors indicate the higher randomized accordance, while the lighter colors indicate the opposite. All taxa names (indicated by the abbreviations consist of the first two letters of the genera and species names, respectively) are listed on top and to the right of the heat map. While cells specify *p*-values > 0.05, indicating that corresponding pairs of nucleotide sequences do not violate the assumption of global stationary, reversibility, and homogeneity conditions.

**Figure 5 insects-12-00570-f005:**
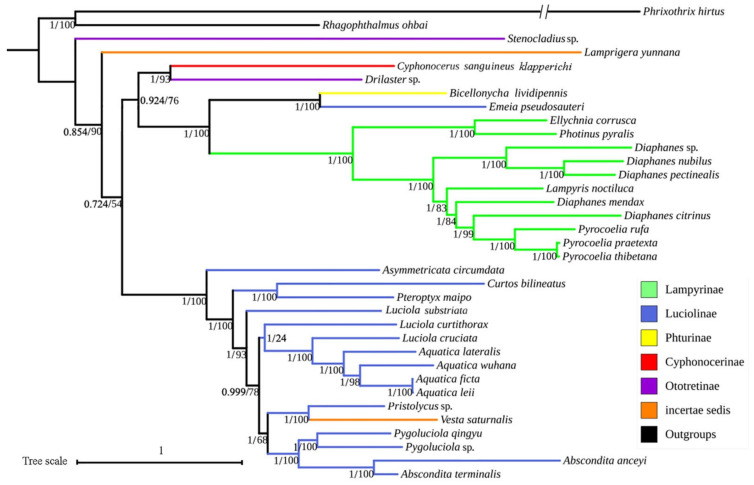
Phylogenetic tree of Lampyridae inferred from the BI analysis of the PCGRNA dataset or ML analysis of PCGrRNA dataset. Numbers under the branches are bootstrap values (right) or posterior probabilities (left).

**Table 1 insects-12-00570-t001:** Information for the representative species’ mitogenomes used for phylogenetic analysis.

Family	Subfamily	Species	Accession Number	Reference
Phengodidae		*Phrixothrix hirtus*	KM923891.1	[54]
Rhagophthalmidae		*Rhagophthalmus ohbai*	NC_010964.1	[55]
Lampyridae	Lampyrinae	*Diaphanes nubilus*	MK292094.1	[2]
		*Diaphanes* sp.	MK292095.1	[2]
		*Diaphanes citrinus*	MK292103.1	[2]
		*Diaphanes mendax*	MK292116.1	[2]
		*Diaphanes pectinealis*	MK292118.1	[2]
		*Ellychnia corrusca*	MG242622.1	Unpublished
		*Lampyris noctiluca*	KX087302.1	[56]
		*Photinus pyralis*	KY778696.1	[57]
		*Pyrocoelia rufa*	AF452048.1	[58]
		*Pyrocoelia praetexta*	MK292115.1	[2]
		*Pyrocoelia thibetana*	MK292117.1	[2]
	Luciolinae	*Abscondita anceyi*	MH020192.1	[59]
		*Abscondita terminalis*	MK292092.1	[2]
		*Aquatica leii*	KF667531.1	[60]
		*Aquatica ficta*	KX758085.1	[61]
		*Aquatica wuhana*	KX758086.1	[61]
		*Aquatica lateralis*	NC_035755.1	[62]
		*Asymmetricata circumdata*	MK292113.1	[2]
		*Curtos bilineatus*	MK292114.1	[2]
		*Emeia pseudosauteri*	MK292112.1	[2]
		*Luciola curtithorax*	MG770613.1	[63]
		*Luciola cruciata*	NC_022472.1	[19]
		*Pristolycus* sp.	MK292099.1	[2]
		*Pteroptyx maipo*	MF686051.1	[64]
		*Pygoluciola qingyu*	MK292093.1	[2]
		*Pygoluciola* sp.	MK292102.1	[2]
		*Luciola substriata*	KP313820.1	[65]
	Incertae sedis	*Vesta saturnalis*	MK292111.1	[2]
		*Lamprigera yunnana*	MK292091.1	[2]
	Photurinae	*Bicellonycha lividipennis*	KJ922151.1	[53]
	Ototretinae	*Drilaster* sp.	MK292100.1	[2]
		*Stenocladius* sp.	MK292101.1	[2]
	Cyphonocerinae	*Cyphonocerus sanguineus klapperichi*	MW365445	In this study

Note: “unpublished“ means the sequence with an accession number of the species could be download from the NCBI, but the publication could not be found.

**Table 2 insects-12-00570-t002:** Nucleotide composition and skewness of mitogenomes of *C. sanguineus*
*klapperichi*.

Regions	Size (bp)	A (%)	C (%)	G (%)	T (%)	AT (%)	GC (%)	AT Skew	GC Skew
Full genome	16,443	42.5	13.9	9.4	34.2	76.7	23.3	0.11	−0.19
PCGs	11,008	41.1	14.6	10.3	34	75.1	24.9	0.09	−0.17
1st codon position	3669	33.2	12.2	15.5	39.1	72.3	27.7	−0.08	0.12
2nd codon position	3669	25.2	14.9	12.3	47.7	72.9	27.2	−0.31	−0.10
3rd codon position	3669	37.8	10.6	9.3	42.4	80.2	19.9	−0.06	−0.07
tRNAs	1432	41.6	12.9	9.6	35.9	77.5	22.5	0.07	−0.15
rRNAs	2034	47	13.1	6.4	33.5	80.5	19.5	0.17	−0.34
CR	1776	45.6	11.6	7.2	35.6	81.2	18.8	0.12	−0.23

## Data Availability

The sequence generated in this study is deposited in GenBank with accession number (MW365445).

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
