# Peer review of "Characterization of the First Complete Mitochondrial Genome of Cyphonocerinae (Coleoptera: Lampyridae) with Implications for Phylogeny and Evolution of Fireflies"

_insects, 2021, doi:10.3390/insects12070570_

Round 1

Reviewer 1 Report

Overall, it is a well written manuscript. Descriptive methods and results, solid discussion. The authors had provided complete mitochondrial genome of the fire flies subfamily Cyphonocerinae, which represents novel data and the first record for this subfamily. Well described and analyzed structure, adequate genes annotations, base composition, substitution and evolutionary rates of the mitochondrial genome  between  Cyphonoceriane. Clear description of similarities/alterations in comparison with other Lampyridae in base composition and genomic structure. Finally, construction of phylogeny tree contributed to confirmation or clarification of monophyily of certain lineages. All figures and structures are of high quality and very informative. My suggestion would be to accept the manuscript for publication in its original form.

Author Response

Thanks very much for your high appreciation for our MS.

Reviewer 2 Report

In general the manuscript is well written and the conclusions seem correct. However, I still strongly recommend either major revision or reject with resubmission. The reason is that the manuscript is full of tables, figures and analyses that are of little value and does not add anything new to science. The results that the authors present is something that was published back in the 1990-ies when the first mt genomes were sequenced. Nowadays reporting these feature tables are redundant. Especially as there is nothing novel with the features of the single mt genome of the they sequenced.

The value lies in the phylogenetic analysis of the firefly radiation and I suggest that the authors rather focus on this instead of reporting features that anyway can be found in the genbank file.

I suggest to redo the phylogenetic analysis. The inclusion of tRNA and rRNA in the phylogenetic data set is not needed. This is due to their highly conserved/highly variable regions and overall short length. There are no good evolutionary models for tRNAs/rRNAs and better to use protein-coding genes where appropriate models have been developed.

The authors should make three data sets with protein-coding genes only. 1) PCG with all 3 codon positions 2) PCG with 1st and 2nd codon positions 3) PCG translated to amino acids. They should also use ModelFinder in IQTREE to find the most suitable model for each data set instead of as in the current analysis use GTR for all.

I am not sure of what the standard is for insect mt genomes on where to start the annotation. For vertebrate mt genomes the annotation start at tRNA-Phe and for other invertebrates at CO1. The authors should rearrange and re-annotate accordingly to insect mt genome standards. Second, what is N and J strand (line 160 and line 161)?

Table 1, list of all analysed species –  this is better placed in supplement

Table 2, annotation of mt genome, either in supplement or take out completely. This can be found in the genbank file. And adjust start to CO1.

Figure 2, A+B does not add much to the story, C+D explain better

Figure 3, can go in supplement

Figure 5, heatmap of sequence divergence, this needs to be explained much better. I do not understand what the main message is of this.

Figure 6, tRNA structures, either supplement or remove. This is standard and not worth reporting and especially not in the main manuscript.

Figure 7, analysis of all CR features, also standard and can go in supplement.

Reviewer 3 Report

The manuscript "Characterization of the first complete mitochondrial genomes of Chyphonocerinae with implications for phylogeny and evolution of fireflies" was overall well written and a sound paper. The authors first describe the first mt genome for C. sanguineus klapperichi and then compare it to other firefly genomes using evolutionary and phylogenetic approaches. My few comments are listed below:

  1. The title makes it sound like multiple genomes were sequenced. I would simply take the 's' off of genomes.
  2. The simple summary is far from simple. I think the authors should re-write it to only convey the main purpose of the study and some key findings in layman terms.
  3. PCG is used in the abstract, but it not defined.
  4. Section 2.2. in the Methods is very hard to follow. If possible it would be nice if that section was expanded and the history described more clearly.
  5. The caption for Fig. 3 is not visible.

Another issue to note is some grammatical errors. I encourage the authors to read the manuscript carefully with particular attention to verb tense and singular/plural nouns.

Author Response

I agree with you for all your corrections and suggestions, and accordingly made some changes in the text, please see the details in the text.
